# StructEval: Benchmarking LLMs' Capabilities to Generate Structural Outputs

Jialin Yang[*][†][1]    Dongfu Jiang[*][†][1 6]    Lipeng He[1]    Sherman Siu[1]    Yuxuan Zhang[7]    Disen Liao[1]    Zhuofeng Li[4]    Huaye Zeng[1]    Yiming Jia[1]    Haozhe Wang[3]    Benjamin Schneider[1]    Chi Ruan[5]    Wentao Ma[1]    Zhiheng Lyu[1]    Yifei Wang[1]    Yi Lu[2]    Quy Duc Do[1]    Ziyan Jiang[1]    Ping Nie[1]    Wenhu Chen[†][1 6]

[1]University of Waterloo, [2]University of Toronto, [3]HKUST, [4]Shanghai University, [5]Independent Contributor, [6]Vector Institute [7] University of British Columbia [*]Equal Contribution

[†]{j586yang, dongfu.jiang, wenhuchen, ping.nie}@uwaterloo.ca

Reviewed on OpenReview: https://openreview.net/forum?id=buDwV7LUA7

## Abstract

As Large Language Models (LLMs) become integral to software development workflows, their ability to generate structured outputs has become critically important. We introduce **StructEval**, a comprehensive benchmark for evaluating LLMs' capabilities in producing both non-renderable (JSON, YAML, CSV) and renderable (HTML, React, SVG) structured formats. Unlike prior benchmarks, StructEval systematically evaluates structural fidelity across diverse formats through two paradigms: **1)** generation tasks, producing structured output from natural language prompts, and **2)** conversion tasks, translating between structured formats. Our benchmark encompasses 18 formats and 44 types of task, with novel metrics for format adherence and structural correctness. Results reveal significant performance gaps—even state-of-the-art models like o1-mini achieve only 75.58 average score, with open-source alternatives lagging approximately 10 points behind. We find generation tasks more challenging than conversion tasks, and producing correct visual content more difficult than generating text-only structures. Please see our project page for more detail https://tiger-ai-lab.github.io/StructEval/.

## 1 Introduction

In recent years, there has been a significant surge in the capabilities of large language models (LLMs) in generating human-like text and performing a wide range of natural language processing tasks. State-of-the-art models like GPT-4o (Hurst et al., 2024), OpenAI o1/o3 (Contributors et al., 2024), and Google's Gemini (Team et al., 2023) have achieved superior performance in knowledge QA (Hendrycks et al., 2020; Wang et al., 2024), instruction-following (Chiang et al., 2024; Zhou et al., 2023), and code generation (Zhuo et al., 2024; Jain et al., 2024).

Despite recent advances, many real-world applications require not only fluency in the content of the output but also precise control over its structure. This includes tasks where the expected output must follow specific formats such as JSON, XML, LaTeX, HTML, or code in frameworks like React or Vue. Additionally, in these tasks, we also want the code to render a page that correctly places elements according to the requirements. These types of structured output are essential in domains like software development, data pipelines, user interface generation, and scientific publishing, where incorrect formatting can lead to disrupted pipelines or non-functional outputs.

However, most existing benchmarks focus on the semantic quality (Wang et al., 2024) or reasoning ability of LLMs (Hendrycks et al., 2021; He et al., 2024), with limited emphasis on their ability to produce

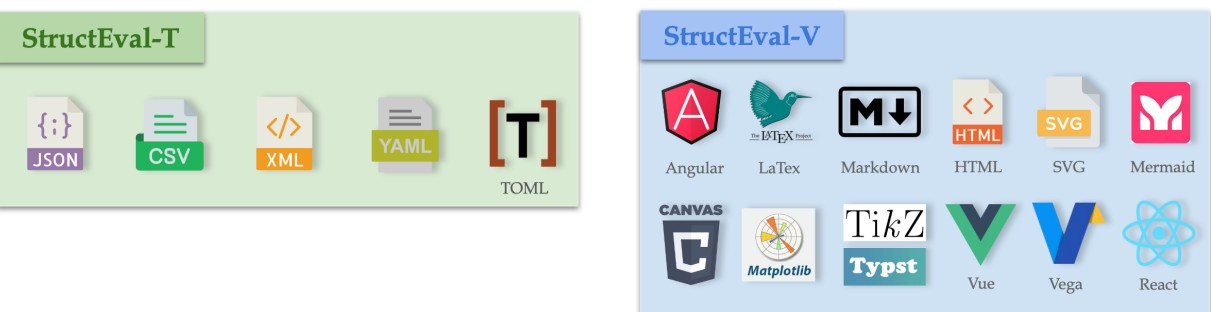

Figure 1: STRUCTEVAL evaluates the LLM's capability to generate structured outputs, including text-only tasks like JSON, TOML, etc, and visual rendering tasks like HTML, React, Latex, etc.

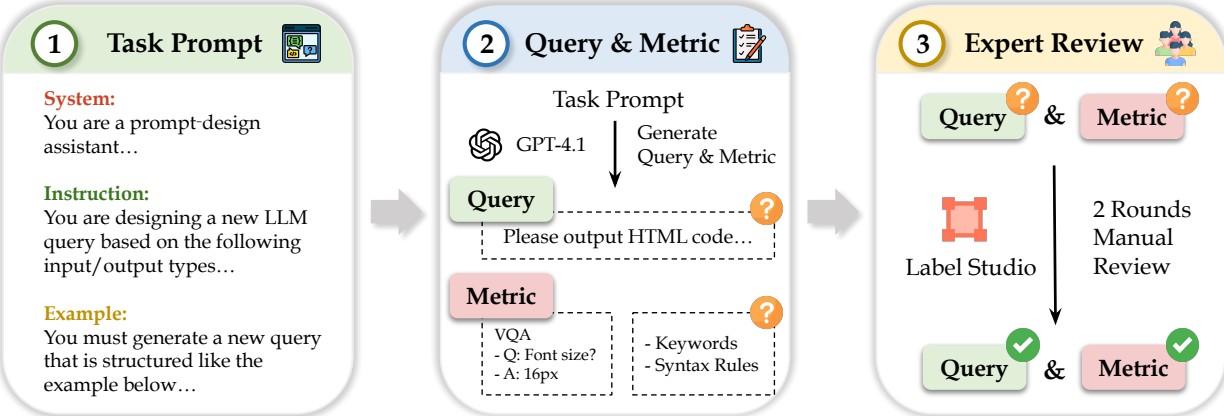

Figure 2: The overall designed annotation pipeline of STRUCTEVAL dataset

format-conforming structured outputs. Some recently proposed benchmarks aim to evaluate the quality of structured outputs tend to target specific modalities, such as code generation (Zhuo et al., 2024) or text-only structures (Gu et al., 2024; Tang et al., 2023), rather than offering comprehensive evaluations across diverse structured formats. As existing benchmarks gradually become more saturated, it is still unknown how the current state-of-the-art models perform in structured generation tasks. We argue that effectively evaluating the models' performance on such tasks is inherently challenging due to the following issues:

**(1) Data Collection Challenges:** Gathering diverse structured tasks and corresponding examples requires domain expertise across multiple formats, with high-quality annotations demanding significant effort and specialized knowledge.

**(2) Evaluation Metric Complexity:** Designing reasonable metrics in a unified form for both text-only structures (JSON, YAML) and visual outputs (HTML, SVG) is difficult, as they require different assessment approaches for structural correctness and visual fidelity.

**(3) Technical Implementation Barriers:** Building a framework that supports execution and evaluation across numerous rendering environments requires complex integration of multiple language interpreters and visualization tools.

To address these challenges, we introduce STRUCTEVAL, a comprehensive benchmark that systematically evaluates LLMs' abilities to produce highly structured output. Our benchmark encompasses 21 distinct formats and 44 task types organized into two complementary subsets: *StructEval-T*, which assesses the generation of text-only structures such as JSON and TOML, and *StructEval-V*, which evaluates the quality of visually rendered outputs from code such as HTML and SVG. Both subsets include generation tasks (converting natural language to structured outputs) and conversion tasks (transforming between two structured

| Subset | # Total Tasks | # Total Examples | # Avg Keywords | # Avg VQA pairs |
|---|---|---|---|---|
| SE-T-gen | 5 | 250 | 7.9 | - |
| SE-T-conv | 14 | 700 | 17.5 | - |
| SE-V-gen | 13 | 650 | 11.1 | 7.9 |
| SE-V-conv | 12 | 435 | 22.2 | 9.0 |
| StructEval | 44 | 2035 | 14.7 | 8.5 |

Table 1: The overall statistics of the STRUCTEVAL dataset. Here "SE" denotes StructEval. "T" and "V" represents the *StructEval-T* and *StructEval-V* subsets respectively. "gen" and "conv" represent the "generation" and "conversion" task types respectively.

formats), See Figure 1 for example formats. To ensure robust evaluation across these diverse formats, we have developed a novel assessment framework that integrates syntactic validity checking, keyword matching, and visual question answering, providing a holistic measure of both structural correctness and output fidelity.

Our comprehensive evaluation reveals significant performance gaps across models and tasks. Even state-of-the-art commercial models like o1-mini achieve only an average score of 75.58, while the best open-source model, such as Llama-3-8B-Instruct, lags 10 points behind, underscoring the performance gap between commercial and open-source LLMs. We observe that generation tasks generally pose greater challenges than conversion tasks, and producing code capable of rendering correct visual content proves more difficult than generating text-only structured outputs. Task difficulty varies considerably across formats: while some tasks are effectively solved by all LLMs with scores exceeding 0.95 (such as Text→Markdown and Text→HTML), others remain particularly challenging with all models scoring below 0.5 (including Text→Mermaid and Matplotlib→TikZ). Through this systematic analysis, we aim to drive progress in structured output generation capabilities that are increasingly crucial for the real-world applications of language models.

## 2 StructEval Dataset

In this section, we first present an overview of our STRUCTEVAL dataset and statistical analysis in subsection 2.1. Next, we elaborate on how we design the whole pipeline for annotation and quality review in subsection 2.2. We will introduce how we design the evaluation metrics for each task in our dataset in section 3.

### 2.1 Overview

As shown in Table 1, our STRUCTEVAL dataset comprises a total of 2,035 examples, covering 44 unique structure generation tasks across 18 structured output formats. The dataset is organized into two main subsets: *StructEval-T* and *StructEval-V*.

- *StructEval-T* is designed to evaluate an LLM's ability to generate structured outputs directly from natural language prompts without rendering. Supported formats include JSON, XML, YAML, Markdown, CSV, TOML, among others. These are highly useful formats in many downstream applications.

- *StructEval-V* assesses an LLM's ability to generate executable code for visual rendering that fulfills a specified visual requirement. This subset includes formats such as HTML, React, Matplotlib, Canvas, LaTeX, SVG, Mermaid, and more. These are widely adopted formats for various applications.

Each example in the dataset is categorized as either *generation* or *conversion*. In *generation* tasks, the model is required to produce structured output based on a natural language description with detailed specifications. In *conversion* tasks, the model must translate structured content from one format to another (e.g., JSON to YAML, HTML to React).

| Rule Type | Example | Description |
|---|---|---|
| Literal key access | `planet.name` | Checks if key `name` exists as a child of object `planet`. |
| Nested lists with index | `planet.moons[0].name` | Verifies first item in `moons` list has a `name` field. |
| Wildcard in lists | `planet.moons.*.name` | Confirms that `name` exists for *any* moon in the list. |
| Backtick quoting | `` data.`key.with.dots` `` | Treats entire quoted token as a single key, useful for special characters. |
| CSV header check | `csv::discovery.location` | Ensures CSV output has a column named `discovery.location`. |
| XML attribute fallback | `@id` | Looks for `id` attribute, using `@` to indicate XML format. |

Table 2: Supported rule types in our path-based evaluation.

---

**StructEval-T Question, KeyWords**

Please output `JSON` code.

**Task:**

Summarize metadata about a fictional scientific article. **Feature Requirements:**

1. Top-level field `"title"` is a string containing the article title.
2. Field `"authors"` is a list of exactly two items.
3. Each element of `"authors"` contains `"name"` (string) and `"affiliation"` (string).
4. Field `"publication.year"` is an integer.
5. Field `"keywords"` is a list of strings.

- - - - - - - - - - - - - - - - - - - - - - - - - - - - - - - - - - - - - - - - - - - - - - - - - - - - - - - - - - - - - - - - - - - - - -

**Keywords:**
- `title`
- `authors[0].name`
- `authors[1].affiliation`
- `publication.year`
- `keywords[2]`

---

Figure 3: Example question and key words of the StructEval-T generation task

| Human Evaluation of VQA Questions | Unfair | Fair | Total | Fair Proportion (%) |
|---|---|---|---|---|
| Correct | 6 | 347 | 352 | **98.58%** |
| Wrong | 39 | 6 | 45 | 13.33% |
| Total | 44 | 353 | 397 | 88.92% |
| **Accuracy (%)** | 13.64% | **98.30%** | 88.66% | |

Table 3: Human evaluation results of sampled VQA questions used in StructEval-V. Each question is annotated as fair or unfair, and correctness is measured by VLM judge performance.

Formally, each example is represented as a triplet $(q, \mathbf{K}, \mathbf{Q^v})$, where $q$ denotes the structure generation question, $\mathbf{K} = \{k_1, \ldots, k_{|\mathbf{K}|}\}$ is a set of keywords expected to appear in the output, and $\mathbf{Q^v} = \{(q_1^v, a_1^v), \ldots, (q_{|\mathbf{Q^v}|}^v, a_{|\mathbf{Q^v}|}^v)\}$ is a set of visual question-answer (VQA) pairs used for evaluating examples in the *StructEval-V* subset(An example StructEval-V task with keywords and VQA pairs is shown in Figure 4). In contrast, for *StructEval-T*, $\mathbf{Q^v}$ is empty and not used during evaluation (An example StructEval-T question and its keywords are shown in Figure 3). To ensure comprehensive evaluation, each example in the dataset contains on average 14.7 keywords and 8.5 VQA pairs, as detailed in Table 1.

To further assess the quality and fairness of the VQA pairs used in *StructEval-V*, we conduct a human expert evaluation. Each VQA question is judged as either *fair*, meaning it can be reasonably answered by a VLM judge using only the rendered image, or *unfair*, typically involving information not visually accessible, such

as precise numeric values or interactive UI elements. Table 3 presents the results of this evaluation. Among 397 sampled VQA pairs, 88.92% were considered fair, and 98.58% of the correct VQA questions were judged fair. Overall, 98.30% of all fair questions could be correctly answered by our VLM judge (GPT-4.1-mini), supporting the validity of our automated evaluation process.

The dataset encompasses a wide spectrum of structured output formats, ranging from widely-used data serialization types like JSON and YAML to visually-renderable formats such as SVG, Mermaid, and TikZ. This diverse format coverage enables a more holistic evaluation of LLMs' capabilities in both structured data modeling and visual code generation. Notably, the inclusion of niche yet expressive formats—such as Typst for typesetting, Mermaid for diagram specification, and TikZ for LaTeX-based graphics—broadens the evaluative scope beyond conventional tasks. These formats collectively span domains including web front-end development, data exchange, scientific visualization, and technical documentation. The distribution of tasks across these formats is shown in Table 7, highlighting the balanced composition of generation and conversion tasks across both textual and visual modalities.

## 2.2 Annotation Pipeline

To construct a high-quality and diverse benchmark, we design a multi-stage annotation pipeline consisting of three key components: 1) task curation, 2) LLM-based synthesis, and 3) expert review (see Figure 2 for an overview of this pipeline). This pipeline ensures both the scalability and accuracy of the STRUCTEVAL dataset.

**Task Prompt**   We begin by identifying a broad spectrum of structure generation and conversion tasks that span both text-based and executable visual formats. These tasks are selected to reflect practical use

---

**StructEval-V Question, Keywords Matching, VQA Pairs**

Please output `HTML` code.

**Task:**

Design a webpage that presents a user's travel itinerary. **Feature Requirements:**

- Include a centered `<h1>` header with the text `"Trip Summary"`.
- Use a `<table>` to list destinations; include 3 rows and 2 columns.
- Apply a class `"highlight"` to the second row.
- Add a `<button>` labeled `"Export PDF"` at the bottom of the page.

- - - - - - - - - - - - - - - - - - - - - - - - - - - - - - - - - - - - - - - - - - - - - - - - - - - - - -

**Keywords:**
- `Trip Summary`
- `highlight`
- `<h1>`
- `Export PDF`

- - - - - - - - - - - - - - - - - - - - - - - - - - - - - - - - - - - - - - - - - - - - - - - - - - - - - -

**VQA Pairs:**
- **Q:** What text is displayed in the `<h1>` header?
  **A:** Trip Summary
- **Q:** How many rows are in the table?
  **A:** 3
- **Q:** What class is applied to the second table row?
  **A:** highlight
- **Q:** What text is on the button at the bottom?
  **A:** Export PDF

Figure 4: Example question, keywords, and VQA pairs for STRUCTEVAL-V generation task

cases and diverse real-world scenarios, covering 18 target formats and 44 distinct task types (also shown in Table 7. Each task specification includes format constraints, input-output expectations, and, where applicable, conversion rules. Please refer to subsection A.4 for a sample task prompt.

**Query/Metric Generation** Given the high cost of fully manual annotation, we leverage a large language model to synthesize an initial pool of candidate examples. Each example consists of a task query and a set of associated evaluation metrics, including keywords for text outputs and visual question-answer (VQA) pairs for visual outputs. This step allows us to rapidly generate a large and varied collection of plausible instances that serve as drafts for human refinement.

**Expert Review** To ensure quality and correctness, we employ a two-pass human review process. Annotators first validate and refine the generated task queries and associated metrics. They are allowed to freely modify, add, or remove any part of the synthesized content to ensure task clarity, completeness, and evaluability. In the second pass, a separate reviewer verifies the consistency and correctness of each example. All annotation is conducted using `LabelStudio` (Tkachenko et al., 2020-2025), an open-source collaborative annotation tool designed for structured data. The final dataset contains 2035 curated examples, carefully reviewed to support robust evaluation across both *StructEval-T* and *StructEval-V* settings.

## 3 StructEval Evaluation

Before the evaluation, we feed the LLM with the questions $q$ in the datasets with the corresponding prompt template defined in Table 4. We require the LLM to output the desired structured outputs between "<|BEGIN_CODE|>" and "<|END_CODE|>" so we can correctly parse the structured outputs for evaluation. For the *StructEval-V*, parsed outputs will be additionally sent to our rendering engines to acquire the rendered visual outputs (see examples in subsection A.3). We then evaluate model outputs using an automatic evaluation pipeline that captures both structural correctness and semantic fidelity. Specifically, we have designed core metrics depending on the task format: **1)** Syntax Score, **2)** Keyword Matching Score, and **3)** Visual Question Answering (VQA) Score.

```
{StructEval Question}

IMPORTANT: Only output the required output format.  You must start the format/code with
<|BEGIN_CODE|> and end the format/code with <|END_CODE|>.  No other text output (explanation,
comments, etc.) are allowed.
Do not use markdown code fences.
```

Table 4: Prompt template used for LLM inference before the evaluation

**Syntax Score.** The Syntax Score verifies the structural correctness of the generated output. For text-based formats such as JSON, YAML, and CSV, this involves parsing the output using a format-specific Python parser. For executable visual formats like HTML, LaTeX, or SVG, the code is rendered using a headless renderer to determine whether it executes successfully. A score of 1 is assigned if the output is syntactically valid or successfully rendered; otherwise, the score is 0. See the subsection A.3 for some correctly rendered images, code produced by the tested LLMs.

**Keyword Matching Score** This metric evaluates whether the generated output contains the required structural elements. Given the reference set of expected keywords $\mathbf{K} = \{k_1, \ldots, k_{|\mathbf{K}|}\}$ for a given task, we assess their presence using exact matching or regular expression rules.

For the tasks of *StructEval-T* such as JSON or XML, keyword matching is performed over field names and values using dot-path references to account for nested hierarchies. The score is computed as the proportion of expected keywords correctly matched in the model's output. Our evaluation supports a variety of path formats as shown in Table 2. The way dot-path rules are created differs depending on the task type.

---

**VQA Prompt Template**

You are given an image and a list of question-answer pairs.

- For each pair, verify if the image content supports the expected answer based on the corresponding question.

- Base your judgment solely on the visual content of the provided image, and the question.

- Do not use any external information or common-sense reasoning beyond what is visible.

- Respond with a JSON object mapping each question number to true or false (e.g., {"1": true, "2": false}).

- If the image is unclear or does not contain enough information to answer, use `null` for that question.

Here are the question-answer pairs: {qa_list}

---

Figure 5: Prompt template used for VQA evaluation. We use GPT-4.1-mini in the benchmark evaluation.

For *generation* tasks, each task prompt includes feature requirements stated in natural language. These requirements define target keys and their relationships to one another (e.g., nesting depth, list membership). Annotators translate each requirement into a concrete dot-path rule using the syntax rules shown in Table 2. For *conversion* tasks, the input is itself a structured format (e.g., YAML or XML). We use an LLM to parse the structural schema of the input—identifying key names, nesting levels, and list structures—and convert them into target dot-path rules that the generated output must preserve.

This approach ensures that models are not only producing syntactically valid outputs, but also preserving the expected structural relationships.

For the tasks of *StructEval-V* such as HTML, and Matplotlib, we simply detect whether the annotated keyword is in the structured outputs and give scores accordingly.

**VQA Score**   This score is used exclusively for tasks in the *StructEval-V* subset, where the output is expected to be visually rendered. After rendering the output, GPT-4.1-mini (Hurst et al., 2024), a vision-language model (VLM), is employed to answer a set of visual questions $\mathbf{Q^v} = \{(q_1^v, a_1^v), \ldots, (q_{|\mathbf{Q^v}|}^v, a_{|\mathbf{Q^v}|}^v)\}$. The VLM will be given both the questions and answers and required to decide whether the VQA pair matches this rendered image. The VQA score is computed as the proportion of correctly answered questions.

Final task scores are calculated as weighted combinations of these metrics, with weights adjusted based on whether the task is renderable. Let $s_s, s_k, s_v \in [0, 1]$ denotes the syntax, keyword matching, and VQA score respectively. For the *StructEval-T* task, the final score $s$ is computed as:

$$s = 0.2 \cdot s_s + 0.8 \cdot s_k \tag{1}$$

For *StructEval-V*, the final score $s$ is computed as:

$$s = 0.2 \cdot s_s + 0.1 \cdot s_k + 0.7 \cdot s_v \tag{2}$$

This evaluation framework provides a unified, fine-grained view of model performance across both structured data generation and visual code synthesis tasks, supporting deeper insights into LLM capabilities across modalities.

## 4 Experiments

### 4.1 Experimental Setup

**Evaluation Models.**   We evaluate a range of open-source and commercial large language models (LLMs) using our benchmark. For open-source models, we use Meta-Llama-3-8B-Instruct Grattafiori et al. (2024), Phi-

3-mini-128k-instruct Abdin et al. (2024a), Phi-4-mini-instruct Abdin et al. (2024b), Qwen2.5-7B-Instruct Yang et al. (2024), and Qwen3-4B Yang et al. (2025). For commercial models, we use Gemini-1.5-pro and Gemini-2.0-flash Team et al. (2023), GPT-4.1-mini and GPT-4o Hurst et al. (2024), GPT-4o-mini, and o1-mini Contributors et al. (2024). All tasks are evaluated in a zero-shot setting using consistent prompts and parameters.

**Inference Setup.** All model generations are performed using `LLM-Engine` Jiang (2024), a unified inference framework that supports both open-source backends (e.g., VLLM, SGLang, Together), and commercial APIs (e.g., OpenAI, Claude, Gemini). For open-source models, we specifically utilize the vLLM engine for efficiency Kwon et al. (2023). For closed source models, we simply call the APIs. As shown in Table 5, we use greedy decoding by default. All tasks are evaluated zero-shot using uniform task prompts defined in Table 4. When performing the VQA evaluation, we select GPT-4.1-mini as the VLM due to its superior multimodal abilities (OpenAI, 2025). We apply the VQA prompt template defined in Figure 5 and ask the VLM to decide whether each VQA pair matches the rendered visual image at once.

| Parameter | Value |
|---|---|
| Max tokens | Unlimited |
| Temperature | 0.0 (deterministic) |
| num_proc | 32 |
| time_out | None |
| num_workers | 5 |
| num_gpu_per_worker | 1 |
| Cache usage | Disabled |
| Batch API | Disabled |
| Hardware | NVIDIA RTX A6000 GPU |

Table 5: Inference configuration

**Evaluation.** Output generations are automatically scored using the evaluation pipeline described in section 3, including syntactic validity checking, keyword matching, and VQA accuracy. GPT-4.1-mini (Hurst et al., 2024) is used as the vision-language model for all VQA-based evaluations.

### 4.2 Main Results

| Models | StructEval-T | | StructEval-V | | Average |
|---|---|---|---|---|---|
| | generation | conversion | generation | conversion | |
| Open Source | | | | | |
| Llama-3.1-8B-Instruct Grattafiori et al. (2024) | 60.22 | 71.26 | 54.44 | 61.15 | 61.77 |
| Meta-Llama-3-8B-Instruct Grattafiori et al. (2024) | 49.18 | 53.65 | 46.61 | 56.91 | 51.59 |
| Phi-3-mini-128k-instruct Abdin et al. (2024a) | 47.39 | 29.78 | 44.77 | 41.23 | 40.79 |
| Phi-4-mini-instruct Abdin et al. (2024b) | 51.38 | 72.39 | 51.62 | 52.48 | 56.97 |
| Qwen2.5-7B-Instruct Team (2024) | 59.21 | 62.18 | 53.28 | 61.43 | 59.03 |
| Qwen3-4B Yang et al. (2025) | **64.95** | **81.13** | **57.00** | **65.08** | **67.04** |
| Close Source | | | | | |
| Gemini-1.5-pro Team et al. (2023) | 88.07 | 74.24 | 58.11 | 66.59 | 71.75 |
| Gemini-2.0-flash Team et al. (2023) | 72.42 | 72.20 | 53.62 | 51.97 | 62.55 |
| GPT-4.1-mini OpenAI (2025) | **92.57** | 75.63 | 64.30 | 70.04 | 75.64 |
| GPT-4o Hurst et al. (2024) | 91.52 | 73.95 | **65.39** | 73.20 | **76.02** |
| GPT-4o-mini Hurst et al. (2024) | 79.86 | 75.57 | 60.77 | **76.54** | 73.19 |
| o1-mini Contributors et al. (2024) | 88.12 | **81.82** | 61.98 | 70.40 | 75.58 |
| Δ (o1-mini - Qwen3-4B) | 23.17 | 0.70 | 4.99 | 5.32 | 8.54 |

Table 6: Main evaluation results of STRUCTEVAL

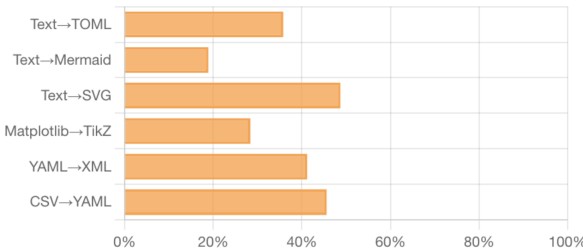 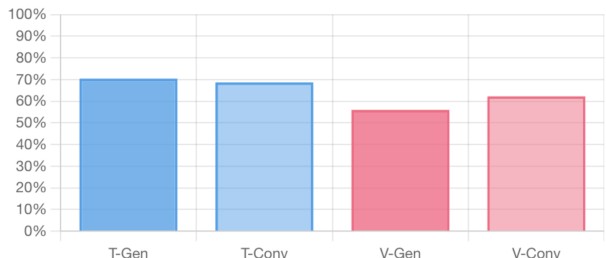

(a) Avg. score over all models based on most challenging subtasks

(b) Avg. score over all models based on the four task types

**Overall Performance**   Table 6 summarizes the performance of all evaluated models across the two main task groups: *StructEval-T* and *StructEval-V*, each further divided into *generation* and *conversion* subtasks. Overall, GPT-4o achieves the highest average score of 76.02% among all 12 models. The best-performing open-source model is Qwen3-4B, with a score of 67.04%, trailing GPT-4o by approximately 10 percentage points. While GPT-4o excels particularly in the *generation* tasks within the *StructEval-V* category, Qwen3-4B demonstrates consistently strong performance across all task types among open-source models. This likely reflects Qwen3-4B's robust reasoning capabilities relative to other open-source alternatives.

In contrast, the lowest-performing model is `phi-3-mini-128k-instruct`, with an average score of only 40.79%. Although one might attribute this to its relatively small size of 3.8 billion parameters, model size alone does not fully explain the poor results. For example, `phi-3-mini` underperforms even compared to similarly sized models such as `phi-4-mini-instruct`. Notably, it achieves the lowest score in *StructEval-T* conversion tasks, a category where models with strong reasoning abilities—such as `o1-mini` (81.82%) and `Qwen3-4B` (81.13%)—tend to perform well.

Error analysis reveals two key failure modes for `phi-3-mini-128k-instruct`. First, in the *TOML-to-YAML* conversion task, the model frequently produces malformed closing tags, outputting `|<|END_CODE|>` instead of the correct `<|END_CODE|>`, which significantly penalizes its score. Second, in the *CSV-to-JSON* conversion task, the model fails to capture hierarchical relationships (e.g., parent-child) specified in the CSV headers, leading to structurally incorrect JSON outputs. These recurring structural errors in *StructEval-T* conversion tasks substantially contribute to the model's overall low performance.

**Open-Source vs. Closed-Source Models**   When comparing open-source models and commercial models, we can see that by $\Delta$ (close$_{avg}$ - open$_{avg}$) value, which is the difference between the average score of commercial source model and open model, that commercial model's score is consistently higher than open-source models, this makes sense given the much larger parameters of commercial models by scaling law. We can see that commercial models exceed open-source models on average the most on generation tasks in StructEval-T setting, and the performance gap is smallest on generation tasks in StructEval-V setting.

**Generation vs. Conversion**   As shown in Figure 6b, a comparison between *generation* and *conversion* tasks in both *StructEval-T* and *StructEval-V* settings reveals that, in general, models perform better on conversion tasks than on generation tasks. An exception to this trend occurs in the *StructEval-T* setting, where commercial models tend to outperform on generation tasks, while open-source models show the opposite behavior—achieving higher scores on conversion tasks.

Under a temperature setting of 1, commercial models attain an average score of 75.78% on *StructEval-T* generation tasks. In contrast, open-source models average only 8.58% on the same tasks for the TOML format. This considerable disparity in TOML generation performance partly explains why commercial models perform better on *StructEval-T* generation tasks overall. However, the performance gap is not confined to TOML—commercial models also lead in the other four generation formats within *StructEval-T*.

In the *StructEval-V* setting, commercial models significantly outperform open-source counterparts on generation tasks involving complex visual formats such as Mermaid and TikZ. These tasks require advanced

visual reasoning capabilities, which are more prevalent in multimodal commercial LLMs like GPT-4o and GPT-4o-mini.

**Subtasks Analysis**   Meanwhile, several tasks in both in generation and conversion types appear to be saturated, with most models achieving scores exceeding 90%. These include generation tasks for common formats such as JSON, HTML, CSV, Markdown, and YAML, as well as conversion tasks like YAML-to-JSON, React-to-HTML, TOML-to-JSON, and Markdown-to-HTML. Such results indicate that LLMs have already mastered many structurally straightforward format transformations.

There remain several challenging tasks where all models struggle significantly (shown in Figure 6a), including generation tasks like Text→TOML, Text→SVG, Text→Mermaid, and Text→Vega, as well as conversion tasks like YAML→XML, CSV→YAML, Matplotlib→TikZ, and Markdown→Angular(see scores in subsection A.2). Both closed-source and open-source models achieve low scores on these tasks, which typically require complex structural or visual reasoning. Notably, the performance gap between closed-source and open-source models is even wider on these challenging subtasks, suggesting that proprietary models may have advantages in handling more complex structural representations and transformation logic.

## 5   Related Work

### 5.1   Large Language Models

Large Language Models (LLMs) have demonstrated remarkable capabilities and gained surging popularity in recent years, ever since the release of ChatGPT (OpenAI, 2023). Over the years, open-source models like Llama (Grattafiori et al., 2024), Phi (Abdin et al., 2024b;a), and Qwen (Yang et al., 2024; 2025) developed by companies like Meta, Microsoft, and Alibaba further facilitated a widespread integration of AI into diverse workflows and everyday applications. Leveraging their large parameter sizes and extensive post-training, LLMs are capable of performing a diverse array of Natural Language Processing (NLP) tasks (Wan et al., 2023). One of the key aspects of the generative capabilities of these models is their ability to generate structured data and transform data from one type to another while maintaining strict adherence to specified formats (Guo et al., 2024). In this paper, we design a new and comprehensive benchmark that evaluates the capability of LLMs to understand, generate, and manipulate structured data across a range of complex, real-world tasks.

### 5.2   Evaluation of LLMs

Evaluating structured output has become a focal point for understanding LLM's limitations (Ning et al., 2025). SoEval (Liu et al., 2024) offers a fast, rule-based check for JSON and XML, but its flat schemas fail to reveal errors in deeper hierarchies. StrucText-Eval (Gu et al., 2024) shifts the task to reasoning over structure-rich text (JSON, YAML, LaTeX) rather than generating the structures themselves, while FOFO (Xia et al., 2024) extends to domains such as law and finance yet covers only a few formats and still relies on human verification. Developer-focused suites like StackEval (Shah et al., 2024) for HTML, CSS, and plotting libraries, and CodeXGLUE (Lu et al., 2021) for multilingual code tasks remain limited to programming artifacts, and Struc-Bench (Tang et al., 2023) concentrates on tabular generation with bespoke metrics. Each benchmark highlights a part of the challenge—be it format adherence, domain coverage, or table fidelity. However, none simultaneously demands broad format coverage, automated grading, and robust transformation capabilities. StructEval addresses these gaps by spanning 18 code and non-code formats, unifying generation, completion, and conversion tasks, and scoring outputs with fully automated structural and vision-based metrics, offering a comprehensive lens on how well LLMs respect and manipulate complex schemas.

### 5.3   Structured Output Generation

The ability to generate structured outputs is central to many real-world applications of LLMs (Gu et al., 2024; Tang et al., 2023). These outputs are not only expected to be semantically coherent but must also adhere strictly to syntactic and structural constraints—violations of which can lead to parsing failures,

rendering errors, or broken downstream applications. Common tasks include generating JSON for API responses (Geng et al., 2025), YAML or TOML for configuration files (Peddireddy, 2024), HTML or React for UI components (Si et al., 2024), and LaTeX or Markdown for technical writing (Wen et al., 2024). Moreover, in data science, models are used to transform unstructured descriptions into structured formats like CSV or tables for integration into analysis pipelines (Li et al., 2023; Su et al., 2024). In publishing and education, tools that convert textual prompts into diagrams (e.g., using TikZ, SVG, or Mermaid) help automate visualization generation (Lee et al., 2025; Rodriguez et al., 2025; Ku et al., 2025). Despite its significance, structured output generation remains challenging due to the need for models to internalize both syntax rules and hierarchical schema relationships across a wide variety of formats. Our STRUCTEVAL first conducts a comprehensive evaluation of existing LLMs on both renderable and non-renderable tasks, showing that they still struggle to correctly generate some data formats including TOML, SVG, and Mermaid.

## 6 Conclusion

In this paper, we have comprehensively studied LLMs' abilities to generate highly structured content. Having the ability to generate fully structured content is highly useful for many downstream tasks. Our paper is among the first few to provide an evaluation suite for that. Our results indicate that current models are still lagging on the renderable structured content, especially on less frequent format. We advocate that the future models should invest more time to optimize their abilities to generate highly structured output.

## Limitations

**Non-interactive formats** Our benchmark focuses on evaluating LLMs' ability to generate static visual rendering formats such as HTML, React, Mermaid, etc. While this approach effectively assesses the model's capacity to produce well-structured and visually coherent outputs, it is currently limited to single-page, non-interactive formats. The evaluation does not account for dynamic behaviors such as button interactions, page transitions, animations, or scroll events, which are essential to many real world user interfaces. Future work could extend the benchmark to include dynamic rendering tasks, enabling a more comprehensive assessment of LLM capabilities in producing fully interactive and responsive user experiences.

**Expert Review** While our dataset underwent a two-pass expert review process to ensure correctness, diversity, and minimize potential biases, the initial content was still generated by large language models. Despite expert oversight, residual biases inherent in the model outputs may persist, particularly in subtle or context-dependent scenarios that are challenging to detect through manual review. Moreover, expert validation, while thorough, may not fully capture the wide range of cultural, social, or contextual sensitivities relevant to diverse user populations. Future work could incorporate broader multi-annotator audits or automated bias detection techniques to further enhance dataset reliability and inclusiveness.

## Acknowledgement

This research was supported in part by the Google Cloud Research Credits Program.

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

# A  Example Appendix

## A.1  Task Distributions

| Subset | Tasks | # Examples |
|---|---|---|
| | **Generation** | |
| StructEval-T | Text → JSON | 50 |
| | Text → CSV | 50 |
| | Text → TOML | 50 |
| | Text → XML | 50 |
| | Text → YAML | 50 |
| StructEval-V | Text → Angular | 50 |
| | Text → Canvas | 50 |
| | Text → HTML | 50 |
| | Text → LaTeX | 50 |
| | Text → Markdown | 50 |
| | Text → Matplotlib | 50 |
| | Text → Mermaid | 50 |
| | Text → React | 50 |
| | Text → SVG | 50 |
| | Text → TikZ | 50 |
| | Text → Typst | 50 |
| | Text → Vega | 50 |
| | Text → Vue | 50 |
| | **Conversion** | |
| StructEval-T | CSV → JSON | 50 |
| | JSON → CSV | 50 |
| | XML → JSON | 50 |
| | JSON → XML | 50 |
| | YAML → JSON | 50 |
| | JSON → YAML | 50 |
| | XML → CSV | 50 |
| | CSV → XML | 50 |
| | XML → YAML | 50 |
| | YAML → XML | 50 |
| | YAML → CSV | 50 |
| | TOML → JSON | 50 |
| | CSV → YAML | 50 |
| | TOML → YAML | 50 |
| StructEval-V | Matplotlib → TikZ | 100 |
| | Markdown → HTML | 50 |
| | HTML → React | 45 |
| | React → HTML | 45 |
| | Vue → HTML | 40 |
| | HTML → Vue | 40 |
| | Markdown → React | 30 |
| | HTML → Angular | 30 |
| | Markdown → Vue | 25 |
| | Vue → React | 15 |
| | Markdown → Angular | 10 |
| | React → Angular | 5 |

Table 7: Statistics of number examples for each task in all the 4 subsets of STRUCTEVAL.

## A.2   Subtask Performance

| Model | T→JSON | T→CSV | T→TOML | T→XML | T→YAML | Avg. |
|---|---|---|---|---|---|---|
| Llama-3.1-8B-Instruct | 78.82 | 81.68 | 6.76 | 59.38 | 74.44 | 60.22 |
| Meta-Llama-3-8B-Instruct | 69.08 | 45.04 | 7.94 | 45.30 | 78.54 | 49.18 |
| Phi-3-mini-128k-Instruct | 68.84 | **93.50** | 0.00 | 37.68 | 36.92 | 47.39 |
| Phi-4-mini-Instruct | 51.50 | 82.56 | 16.12 | 40.20 | 66.54 | 51.38 |
| Qwen-2.5-7B-Instruct | 84.40 | 90.62 | 13.22 | 61.30 | 46.52 | 59.21 |
| Qwen-3-4B | 90.96 | 76.44 | 7.44 | 71.16 | 78.74 | 64.95 |
| Gemini-1.5-pro | 94.06 | **100.00** | 75.38 | 73.32 | **97.58** | 88.07 |
| Gemini-2.0-flash | 48.88 | 98.40 | 78.78 | 44.60 | 91.44 | 72.42 |
| GPT-4.1-mini | **99.26** | 99.92 | **91.34** | **77.06** | 95.26 | **92.57** |
| GPT-4o | **99.36** | **100.00** | 90.22 | 70.32 | **97.68** | 91.52 |
| GPT-4o-mini | 97.88 | 99.90 | 29.56 | 75.10 | 96.84 | 79.86 |
| o1-mini | 92.56 | 99.24 | 89.40 | 71.12 | 88.28 | 88.12 |

Table 8: StructEval-T Generation Scores

| Model | T→Ang. | T→LaTeX | T→MD | T→MPL | T→React | T→SVG | T→TikZ |
|---|---|---|---|---|---|---|---|
| Llama-3.1-8B-Instruct | 61.22 | 78.04 | 87.34 | 80.52 | 64.30 | 44.18 | 46.92 |
| Meta-Llama-3-8B-Instruct | 48.92 | 68.40 | 72.06 | 56.54 | 55.24 | 40.16 | 28.04 |
| Phi-3-mini-128k-Instruct | 48.28 | 63.88 | 64.16 | 59.38 | 44.12 | 35.78 | 32.44 |
| Phi-4-mini-Instruct | 62.60 | 72.92 | 88.90 | 71.30 | 58.46 | 39.72 | 35.28 |
| Qwen-2.5-7B-Instruct | 63.08 | 66.68 | 81.02 | 74.70 | 65.48 | 47.30 | 48.88 |
| Qwen-3-4B | 48.80 | 72.60 | **92.80** | 89.54 | **77.06** | 53.44 | 55.38 |
| Gemini-1.5-pro | **90.62** | 76.94 | 94.00 | 84.96 | 33.68 | 54.72 | 69.44 |
| Gemini-2.0-flash | 44.28 | 75.26 | 92.06 | 75.34 | 46.64 | **56.72** | 61.24 |
| GPT-4.1-mini | 84.52 | 76.20 | 91.80 | **96.34** | 69.58 | 58.74 | 69.74 |
| GPT-4o | 87.42 | 75.18 | 93.02 | 95.76 | 74.66 | 56.78 | 62.32 |
| GPT-4o-mini | 86.72 | **78.44** | 94.36 | 95.36 | 75.46 | 53.98 | 60.76 |
| o1-mini | 89.30 | 49.24 | 92.08 | 96.06 | 71.98 | 58.12 | **71.86** |

Table 9: StructEval-V Generation Scores (Part 1)

| Model | T→HTML | T→Mermaid | T→Typst | T→Vega | T→Vue | T→Canvas | Avg. |
|---|---|---|---|---|---|---|---|
| Llama-3.1-8B-Instruct | 95.96 | 9.02 | 23.38 | 28.36 | 57.90 | 30.56 | 54.44 |
| Meta-Llama-3-8B-Instruct | 72.52 | 6.04 | 29.46 | 30.74 | **66.50** | 31.28 | 46.61 |
| Phi-3-mini-128k-Instruct | 92.10 | 11.12 | 22.90 | 35.56 | 39.84 | 32.50 | 44.77 |
| Phi-4-mini-Instruct | 97.24 | 9.30 | 42.22 | 34.72 | 29.48 | 28.90 | 51.62 |
| Qwen-2.5-7B-Instruct | 92.92 | 6.16 | 33.44 | 30.56 | 37.90 | 44.52 | 53.28 |
| Qwen-3-4B | 98.80 | 13.62 | 9.92 | 45.28 | 29.42 | **54.28** | 57.00 |
| Gemini-1.5-pro | **99.30** | 15.94 | 11.60 | 65.18 | 29.66 | 29.36 | 58.11 |
| Gemini-2.0-flash | 99.26 | 9.66 | **45.28** | 29.74 | 32.46 | 29.16 | 53.62 |
| GPT-4.1-mini | 99.30 | **43.46** | 9.96 | 48.28 | 38.44 | 49.60 | 64.30 |
| GPT-4o | 99.22 | 36.00 | 23.94 | **72.20** | 40.04 | 33.54 | **65.39** |
| GPT-4o-mini | 99.02 | 30.50 | 9.96 | 41.28 | 33.66 | 30.50 | 60.77 |
| o1-mini | **99.44** | 27.76 | 9.98 | 65.68 | **40.76** | 33.52 | 61.98 |

Table 10: StructEval-V Generation Scores (Part 2)

| Model | C→JSON | J→CSV | X→JSON | J→XML | Y→JSON | J→YAML | X→CSV |
|---|---|---|---|---|---|---|---|
| Llama-3.1-8B-Instruct | 34.14 | 95.96 | 68.62 | 56.02 | 94.00 | 92.52 | 98.98 |
| Meta-Llama-3-8B-Instruct | 31.40 | 48.00 | 69.24 | 55.40 | 90.00 | 74.00 | 48.26 |
| Phi-3-mini-128k-Instruct | 24.88 | 87.28 | 8.00 | 12.40 | 23.20 | 32.80 | 33.92 |
| Phi-4-mini-Instruct | 45.42 | 97.62 | 89.56 | 61.90 | **100.00** | **100.00** | 90.70 |
| Qwen-2.5-7B-Instruct | 31.36 | 95.74 | 33.14 | 31.04 | 50.00 | 95.24 | 77.72 |
| Qwen-3-4B | 55.28 | **100.00** | **92.84** | 65.98 | **100.00** | 98.00 | **99.78** |
| Gemini-1.5-pro | 48.14 | **100.00** | 40.14 | 67.14 | 98.00 | **100.00** | **99.78** |
| Gemini-2.0-flash | 25.72 | **100.00** | 32.60 | **69.76** | **100.00** | **100.00** | **99.78** |
| GPT-4.1-mini | **55.52** | **100.00** | 38.68 | **69.76** | **100.00** | **100.00** | **99.78** |
| GPT-4o | 38.56 | 99.74 | 66.46 | **69.76** | **100.00** | **100.00** | **99.78** |
| GPT-4o-mini | **58.52** | **100.00** | 73.26 | 65.98 | 98.00 | **100.00** | 98.22 |
| o1-mini | 58.46 | **100.00** | 82.70 | 68.60 | **100.00** | **100.00** | **99.78** |

Table 11: StructEval-T Conversion Scores (Part 1)

| Model | C→XML | X→YAML | Y→XML | Y→CSV | Toml→JSON | C→YAML | Toml→YAML | Avg. |
|---|---|---|---|---|---|---|---|---|
| Llama-3.1-8B-Instruct | 20.20 | 86.96 | 39.90 | 88.32 | 86.90 | 49.54 | 85.62 | 71.26 |
| Meta-Llama-3-8B-Instruct | 17.28 | 54.48 | 38.12 | 61.90 | 63.38 | 36.50 | 63.18 | 53.65 |
| Phi-3-mini-128k-Instruct | 9.50 | 20.56 | 22.42 | **87.58** | 8.80 | 19.10 | 26.46 | 29.78 |
| Phi-4-mini-Instruct | 21.72 | 60.00 | 48.28 | 84.14 | 86.02 | 66.22 | 61.84 | 72.39 |
| Qwen-2.5-7B-Instruct | 18.12 | 81.62 | 24.16 | **97.62** | 78.22 | 70.86 | 85.68 | 62.18 |
| Qwen-3-4B | 24.82 | **94.10** | 48.68 | **98.94** | 96.92 | 65.08 | **95.36** | 81.13 |
| Gemini-1.5-pro | 27.14 | 42.96 | 47.56 | **100.00** | 99.76 | 71.40 | 97.36 | 74.24 |
| Gemini-2.0-flash | 17.74 | 59.02 | 46.36 | **100.00** | 99.26 | 63.18 | 97.36 | 72.20 |
| GPT-4.1-mini | **29.36** | 59.18 | 48.36 | **100.00** | **100.00** | 60.82 | 97.36 | 75.63 |
| GPT-4o | 27.40 | 44.28 | 48.76 | **100.00** | **100.00** | 43.20 | 97.36 | 73.95 |
| GPT-4o-mini | 29.62 | 40.20 | **48.76** | 98.10 | **100.00** | 50.00 | 97.36 | 75.57 |
| o1-mini | 29.26 | 88.62 | 48.36 | **100.00** | **100.00** | **72.40** | 97.36 | 81.82 |

Table 12: StructEval-T Conversion Scores (Part 2)

| Model | R→HTML | V→HTML | MD→React | HTML→Ang. | MD→Vue | MPL→TikZ |
|---|---|---|---|---|---|---|
| Llama-3.1-8B-Instruct | 88.36 | 84.65 | 43.23 | 60.90 | 36.36 | 16.26 |
| Meta-Llama-3-8B-Instruct | 86.82 | 85.23 | 33.73 | 52.83 | 29.52 | 8.29 |
| Phi-3-mini-128k-Instruct | 70.73 | 73.85 | 30.80 | 32.77 | 27.32 | 17.15 |
| Phi-4-mini-Instruct | 92.27 | 81.82 | 28.50 | 33.47 | 33.88 | 15.70 |
| Qwen-2.5-7B-Instruct | 89.29 | 79.53 | 34.70 | 68.67 | 33.80 | 26.32 |
| Qwen-3-4B | **95.53** | 89.65 | 54.23 | 55.10 | 34.64 | 25.64 |
| Gemini-1.5-pro | 95.24 | **91.27** | 34.83 | **86.43** | 30.96 | 38.82 |
| Gemini-2.0-flash | 93.02 | 88.67 | 32.37 | 29.30 | 32.00 | 17.46 |
| GPT-4.1-mini | 95.22 | 90.12 | 52.87 | 81.97 | 31.96 | 36.80 |
| GPT-4o | 95.36 | 90.55 | 74.20 | 87.17 | **37.56** | 39.69 |
| GPT-4o-mini | 95.07 | **91.58** | 80.40 | 87.73 | 31.96 | **42.47** |
| o1-mini | 95.09 | 89.65 | 58.37 | 87.90 | 36.80 | 40.60 |

Table 13: StructEval-V Conversion Scores (Part 1)

| Model | MD→HTML | HTML→React | HTML→Vue | V→React | MD→Ang. | R→Ang. | Avg. |
|---|---|---|---|---|---|---|---|
| Llama-3.1-8B-Instruct | 88.28 | 55.02 | 72.93 | 75.73 | 26.90 | 85.20 | 61.15 |
| Meta-Llama-3-8B-Instruct | 84.52 | 73.91 | 75.28 | 62.73 | 33.10 | 57.00 | 56.91 |
| Phi-3-mini-128k-Instruct | 65.60 | 42.16 | 34.65 | 33.00 | 25.10 | 41.60 | 41.23 |
| Phi-4-mini-Instruct | 92.44 | 57.11 | 41.05 | 55.87 | 26.50 | 71.20 | 52.48 |
| Qwen-2.5-7B-Instruct | 85.16 | 69.20 | 80.02 | 50.87 | 35.00 | 84.60 | 61.43 |
| Qwen-3-4B | 90.20 | 65.31 | **83.05** | 68.13 | 34.50 | 85.00 | 65.08 |
| Gemini-1.5-pro | 95.28 | 40.62 | **86.65** | 64.00 | **49.80** | 85.20 | 66.59 |
| Gemini-2.0-flash | **96.60** | 41.04 | 67.77 | 68.00 | 28.20 | 29.20 | 51.97 |
| GPT-4.1-mini | 96.40 | **88.09** | 46.28 | **86.47** | 49.10 | 85.20 | 70.04 |
| GPT-4o | 95.32 | 88.31 | 62.55 | 78.93 | 48.20 | 80.60 | 73.20 |
| GPT-4o-mini | 93.14 | **88.42** | 79.75 | **81.20** | 49.20 | **97.60** | **76.54** |
| o1-mini | 94.48 | 72.18 | 77.77 | 65.60 | 41.20 | 85.20 | 70.40 |

Table 14: StructEval-V Conversion Scores (Part 2)

* T - Text, C – CSV, J – JSON, X – XML, Y – YAML, Ang. – Angular, MD – Markdown, MPL – Matplotlib, R – React, V – Vue.

## A.3 Examples of rendered image

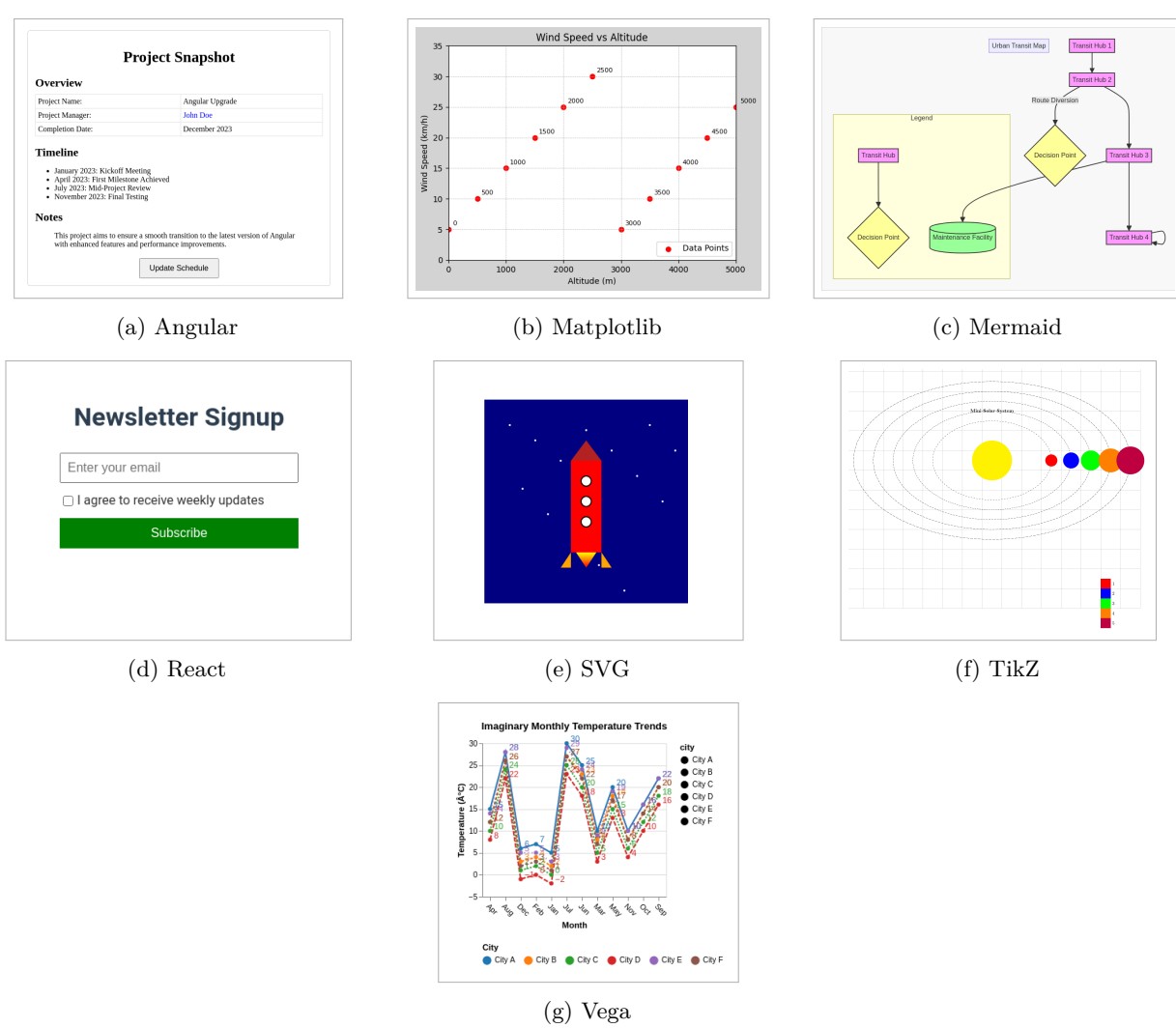

(a) Angular

(b) Matplotlib

(c) Mermaid

(d) React

(e) SVG

(f) TikZ

(g) Vega

Figure 7: Example images rendered in STRUCTEVAL tasks.

## A.4 Task Generation Prompt

---

**Sample Prompt**

You are a prompt-design assistant building benchmark items for conversion tasks.
**Input Format:** {input_type}
**Output Format:** {output_type}
**Your task:** Think silently through the checklist and then output a single JSON object with:

- `"raw_output_metric"`: dot-paths for the expected keys/attributes in the {output_type} structure
- `"query"`: A generated input format {input_type} code inside `...` tags.

**Assumed Mapping Rule (state it implicitly in the paths):**

- **No XML attributes** unless absolutely necessary.
  If an attribute is required, map it to a key prefixed with `"@"`, and include that in dot-paths.

---

**CHECKLIST (INTERNAL – DO NOT OUTPUT)**

1. Pick a super creative and random domain.
2. Generate {input_type} code with:
     - At least two levels of nesting
     - At least one list inside an object/element
3. Avoid XML attributes where possible; prefer child elements.
4. Wrap the code in `...` tags.
5. Dot-path rules:
     - JSON / YAML / TOML: `parent.child`, `list[0].child`
     - XML: `element.child` or `element.@attr` (only if used)
     - CSV: `csv::Header` (not used here)

---

**OUTPUT FORMAT**

```
{
  "raw_output_metric": ["<dot_path1>",
                        "<dot_path2>", ...],
  "query": "..."
}
```

---

Figure 8: Example task generation prompt

