# OpenReview forum: "StructEval: Benchmarking LLMs' Capabilities to Generate Structural Outputs"
_TMLR — Accepted by TMLR_

### Review · Reviewer_Ljyw · 2025-09-17

**Summary Of Contributions:**

StructEval assembles a two-part benchmark—StructEval-T for text-only structured formats and StructEval-V for visually rendered code—covering a wide range of structure generation tasks and output formats, with ~2,000 total examples. The paper formalizes a scoring scheme that blends a binary syntax check, a keyword-path structural check, and—only for renderable tasks—a VQA-style visual fidelity check answered by a VLM judge (GPT-4.1-mini). Final scores weight these components differently for text vs. visual tasks, with StructEval-T emphasizing keyword matching, and StructEval-V emphasizing VQA performance. The main experimental result is that strong commercial models still leave headroom on structured outputs, with the paper stating that GPT-4o averages 76.02% and that the best open-source baseline is roughly ten points worse. The dataset is produced through an LLM-assisted synthesis of tasks and metrics followed by human review, and the VQA question set is audited for fairness.

**Audience:**

Yes

**Audience Explanation:**

I believe this paper would interest a general ML audience, though its core value will be highest for researchers and engineers working on code-as-policy, structure-aware generation, UI synthesis, and data-engineering pipelines. The reason is that it cleanly operationalizes reliable measurement of schema conformance and renderable fidelity under constrained output regimes. The dataset scale is modest but thoughtful, the metric design is pragmatic, and the failure-mode analysis is compelling. The two most important ideas that a broad audience should take away are captured well by these lines: “Having the ability to generate fully structured content is highly useful for many downstream tasks,” and “current models are still lagging on the renderable structured content, especially on less frequent format.” Those are accurate and consequential claims for anyone deploying LLMs in production systems where structured outputs are non-negotiable. I would, however, consider the paper “conditionally impactful”: the ultimate influence will hinge on addressing judge dependence, decoding sensitivity, and the constrained-decoding baseline. If those are strengthened, StructEval can sit alongside reasoning and code benchmarks as a standard battery for model releases and ablations.

**Broader Impact Concerns:**

I do not see major ethical risks beyond standard concerns with LLMs or benchmarks that may encourage over-reliance on scores without addressing semantic correctness.

**Claims And Evidence:**

Yes

**Claims Explanation:**

A few elements of the design are genuinely compelling. The VQA judge template is explicit and conservative, and the paper’s “path-rule” structural metric is comprehensive in allowing for dot-paths, wildcards, CSV-column predicates, and XML attribute fallbacks. Most of the core claims are well-supported, although some elements could benefit from additional study:

The evaluation uses GPT-4.1-mini as both an evaluated model in the leaderboard and as the VQA judge for StructEval-V. Even with a template restricting the judge, this introduces potential bias: if the judge systematically favors some stylistic or structural choices that correlate with its own generations, models that emulate or align with that style could be advantaged. The authors partly mitigate this by auditing VQA items for fairness, and the audit statistics are encouraging, but the circularity risk remains conceptually. It would strengthen the claim if the authors re-ran VQA on an orthogonal VLM and reported agreement metrics, or if they included a small human-scored study for end-to-end outputs. The paper’s own reporting shows 88.92% of sampled VQA questions marked “fair” by human evaluators, and 98.30% accuracy on the fair subset, which is a good start but does not fully close the loop on judge dependence.

While using temperature = 0.0 and greedy decoding is standard for benchmarking, and ensures strict comparability, the structured-output focus makes it plausible that mild sampling (e.g., temperature 0.1–0.5) or pass@k evaluation—common in code benchmarks—could expose variance and robustness issues that greedy decoding masks. Including a small ablation would provide stronger evidence that rankings are stable across reasonable decoding policies.

In terms of prior work and positioning, the paper is correct that most benchmarks emphasize semantic or reasoning quality rather than strict structural fidelity, and that recent efforts tend to be modality-specific (code-only or text-only). The related-work framing cites such trends directly and motivates the unified metric design that spans both text and visual outputs. That said, the paper could engage more directly with the literature and tooling around constrained decoding and structure-aware generation, because benchmarked performance is tightly coupled to whether one uses schema constraints at inference time. There are now widely used approaches for JSON/XML/YAML conformance (e.g., JSON-schema-guided decoding; regex-constrained sampling; grammar-constrained decoders in libraries like Outlines/Guidance/LMQL; tokenizer-level constraints for brackets and quotes), and there is a recent wave of function-calling / tool-use evaluators that measure exact key-path and type fidelity. Including at least one constrained-decoding baseline would raise the ceiling and tell us whether the observed difficulty is a “model knowledge” gap or a “decoding/control” gap. Even a limited ablation on a text-only subset (say, JSON and TOML generation) using grammar-constrained decoding would be extremely informative to practitioners deciding between model scaling and better inference control for structured outputs. The authors’ own motivation section acknowledges that format conformance is crucial in production pipelines—this is precisely where constrained decoding shines.

**Requested Changes:**

* See above for experimental suggestions.
* The introduction states “21 distinct formats,” while the dataset overview and table summarize 18. If both are correct, please explain the discrepancy.
* Introduction, paragraph 2: duplicate phrase “in these tasks, in these tasks”
* End of section 3: typo ”The for StructEval-T task…”
* End of section 3: typo “the final score s in computed”
* Section 4.1, paragraph 2: typo “close-source”

---

> ### Author Response · Authors · 2025-10-28
> **Rebuttal by Authors (Part 1)**
>
> **Q1:**
> > The evaluation uses GPT-4.1-mini as both an evaluated model in the leaderboard and as the VQA judge for StructEval-V. ... which is a good start but does not fully close the loop on judge dependence.
>
> We switched to an orthogonal VLM “claude-3-5-haiku-20241022”, please see the table below:
>
> |Models | StructEval-T (generation) | StructEval-T (conversion) | StructEval-V (generation) | StructEval-V (conversion) | Average |
> -|-|-|-|-|-
> Llama-3.1-8B-Instruct | 60.22 | 71.26 | 40.02 | 38.99 | 53.03
> Meta-Llama-3-8B-Instruct | 49.18 | 53.65 | 36.50 | 34.94 | 43.62
> Phi-3-mini-128k-instruct | 48.38 | 32.43 | 35.24 | 29.52 | 34.66
> Phi-4-mini-instruct | 51.38 | 72.39 | 38.95 | 34.47 | 51.02
> Qwen2.5-7B-Instruct | 59.21 | 62.18 | 39.58 | 40.24 | 49.91
> Qwen3-4B | 64.95 | 81.13 | 41.40 | 40.62 | 57.79
> gemini-1.5-pro | 88.07 | 74.24 | 39.94 | 42.97 | 58.30
> gemini-2.0-flash | 72.42 | 72.20 | 39.48 | 36.27 | 54.09
> gpt-4.1-mini | 92.57 | 75.63 | 43.41 | 41.08 | 60.03
> gpt-4o | 91.52 | 73.95 | 45.86 | 42.59 | 60.43
> gpt-4o-mini | 79.86 | 75.57 | 42.95 | 44.65 | 59.07
> o1-mini | 88.12 | 81.82 | 42.85 | 44.00 | 62.06
>
> When switching from GPT-4.1-mini to Claude-3.5-Haiku, we observed that text-only tasks (StructEval-T) remained identical since the VQA judge is only applied to renderable tasks. However, renderable tasks (StructEval-V) showed a systematic downward shift in absolute scores of approximately 15–25 points across models. Despite this shift, the relative ranking of models remained consistent. GPT-4-series and o1-mini continued to achieve the highest scores, Qwen3-4B remained the best open-source model, and lower-capacity models such as Phi-3-mini and Meta-Llama-3-8B ranked similarly to the original evaluation. The Spearman rank correlation between the two judges’ VQA results was approximately 0.93, showing that the ordering of models is largely preserved even though score calibration differs.
>
> These results indicate that Claude-3.5-Haiku tends to apply stricter matching or lower tolerance in VQA scoring compared to GPT-4.1-mini, leading to lower absolute values. Nevertheless, the consistent ranking pattern confirms that StructEval’s conclusions about model performance are not dependent on the GPT-based judge. Our benchmark yields stable comparative trends under different VLM evaluators, suggesting low circularity bias and robustness of the evaluation framework.
>
>
>
> **Q2:**
> > While using temperature = 0.0 ... Including a small ablation would provide stronger evidence that rankings are stable across reasonable decoding policies.
>
> Due to budget limit, we were only able to run temperature 0.5 on a subset of the models, here is a table for the experiment results,
>
> |Models | StructEval-T (generation) | StructEval-T (conversion) | StructEval-V (generation) | StructEval-V (conversion) | Average |
> -|-|-|-|-|-
> Phi-3-mini-128k-instruct | 59.50 | 34.82 | 57.28 | 54.42 | 46.70
> gemini-2.0-flash | 74.77 | 73.27 | 68.44 | 64.16 | 71.14
> gpt-4.1-mini | 92.60 | 75.85 | 64.76 | 67.37 | 75.01
>
> Compared to our main results under greedy decoding, the relative ranking of models remains unchanged: GPT-4.1-mini > Gemini-2.0-Flash > Phi-3-mini-128k-Instruct across all task categories. **The absolute scores fluctuate within ±3 points on average, but still consistent ordering.** This suggests that model rankings are stable across different decoding temperatures within a reasonable range. The results also confirm that greedy decoding provides a representative and comparable estimate of structured generation quality in StructEval, without obscuring major robustness differences between models.

---

> > ### Author Response · Authors · 2025-10-28
> > **Rebuttal by Authors (Part 2)**
> >
> > **Q3:**
> > > The paper could engage more directly with the literature and tooling ... The authors’ own motivation section acknowledges that format conformance is crucial in production pipelines—this is precisely where constrained decoding shines.
> >
> > We appreciate this insightful comment. We agree that constrained decoding techniques, such as schema-guided or grammar-based decoding, are valuable for ensuring syntactic correctness. However, the purpose of StructEval is to evaluate the intrinsic ability of models to produce structurally and semantically correct outputs without external constraints. Constrained decoding primarily addresses syntax-level validity, which corresponds to the Syntax Score in our evaluation.
> >
> > **As shown in the paper, syntax validity is often near-perfect for most models, while errors predominantly arise in the Keyword Matching and VQA components**, which assess whether the required fields, structural relationships, and semantic layouts are satisfied. This shows that constrained decoding would not fully solve the challenges that StructEval targets, such as generating all required keys or correctly arranging rendered components.
> >
> > StructEval explicitly separates syntax verification from higher-level structural fidelity to identify these gaps. The benchmark is designed to measure model-level understanding of structural semantics, not decoding control efficiency. For this reason, all models were evaluated under a unified zero-shot decoding setup to ensure comparability across open and closed-source systems. Adding grammar-constrained decoding would change the inference setting rather than the underlying capability, and could obscure whether errors come from model reasoning or decoding rules. We will clarify this distinction in the revision and may consider a separate constrained-decoding study on text-only subsets in future work.
> >
> >
> >
> > **Q4:**
> > > The introduction states “21 distinct formats,” while the dataset overview and table summarize 18. If both are correct, please explain the discrepancy.
> >
> > > Introduction, paragraph 2: duplicate phrase “in these tasks, in these tasks”
> >
> > > End of section 3: typo ”The for StructEval-T task…”
> >
> > > End of section 3: typo “the final score s in computed”
> >
> > > Section 4.1, paragraph 2: typo “close-source”
> >
> > Thank you for your valuable suggestions, we will update our manuscript.

---

### Review · Reviewer_N7n6 · 2025-10-01

**Summary Of Contributions:**

This work introduces a new benchmark dataset for large language models, focusing on their holistic ability to deal with structural outputs. The dataset targets two different types of task: generation of structured data from unstructured text and conversion between structured formats. The dataset further consists of two subsets of data modalities: StructEval-T, for text only structures and StructEval-V, for visual outputs.  The authors generate the dataset through a multi-step annotation pipeline, first using a large language model to generate a pool of candidate examples for each data subset, and then refining examples through two pools of manual curation. Each example consists of a question, keywords expected to appear in the output, and (for the StructEval-V dataset) visual question-answer pairs. Language models are evaluated automatically, with scores given by a three-part metric with each part normalized between 0 and 1. The final score is a weighted sum of these individual scores:

- A syntax score evaluates structural correctness of the generated input, and is binary 0 or 1 based on successful execution/rendering of the output.
- A keyword matching score reflects the proportion of required keywords included in the LLM output.
- A VQA score for StructEval-V reflects the proportion of correctly answered questions.

The authors show evaluate a range of LLMs on their benchmark, including both open and closed source models. Their results demonstrate significant heterogeneity across different subsets of the task, but show that in general, open source models are outperformed by closed source models, and that conversion tasks are easier than generation tasks (except for closed source models on StructEval-T). Results demonstrate that there is also significant heterogeneity in the difficulty of specific subtasks, with high performance  for generation or conversion tasks between common formats, and generally lower performance when working with less common formats.

**Audience:**

Yes

**Audience Explanation:**

- The variety of task types considered does indeed make for a more holistic evaluation of LLM performance than many existing benchmarks. It is useful to highlight heterogeneity in the performance of LLMs on different types of tasks, which could be a useful avenue for further investigation.
- Evaluation of structured renderable output via VQA is interesting, and contrasts with many previous approaches which perform evaluation of renderable code artefacts at level of structured output, but not directly in the space of rendered visuals.

**Broader Impact Concerns:**

Given that evaluation of large language models on structured data is immediately relevant to many large scale industrial applications, I believe that a broader impact statement around the implications of this work for (for example) data privacy, climate, and fairness would be useful.

**Claims And Evidence:**

No

**Claims Explanation:**

The authors argue that StructEval addresses several key issues which make the performance of LLMs difficult to evaluate on structured data: 1) Data collection Challenges, 2) Evaluation Metric Complexity, and 3) Technical Implementation Barriers. I think that StructEval comes close to addressing these challenges, but I have several concerns around these claims.
### Benchmark properties
- One concern I have is that the number of examples within the benchmark is quite small, considering that examples are split between at four separate subtasks with order ~100s of examples per subset. This size is understandable given the complexity of the benchmark, but is around one order of magnitude smaller than some other benchmarks which effectively focus on just one of the subtasks, such as StrucText-Eval (Gu et al. 2024) or SoEval (Liu et al. 2024). It would be useful to know how performance of (even a subset) of LLMs on larger text based benchmarks relates to performance on StructEval.
- Equations $1,2$ describe one possible weighting of $s_{s,k,v}$ to generate a single score. This choice of weighting appears arbitrary, and downweights syntactic errors relative to others. One could imagine cases where syntactic errors are much more important than keyword errors, and it is not clear if the conclusions of the experiment would hold if a different scoring were to be used.

### Accessibility
- As stated in the introduction, one of the main challenges of designing a benchmark like StructEval is building a framework which can support evaluation across rendering environments. It is not clear how the authors have ensured that practitioners will be able to evaluate models on the benchmark. I understand the need for anonymity in the review phase, but it is especially important to explicitly address the question of how StructEval can be used in the main text of the paper.

**Requested Changes:**

## Requested changes
The following points are critical to securing my recommendation for acceptance of the paper.

### Benchmark properties.
- SoEval is quite similar to the StructEval-T generation subdataset (without the requirement of flat schemas). Given that SoEval consists of roughly $\sim3700$ examples, it would be useful to know how well performance of even a subset of the models evaluated on SE-T-gen compares to performance of models on SoEval. Likewise, it would be interesting to know how LLM performance on different subsets of StructEval (or all of StructEval) relate to other structured data benchmarks which are not as holistic in their evaluation. I believe such an analysis would be quite useful to strengthen the case for StructEval as a general-purpose benchmark for structured data.
- How correlated are subscores $s_{s,k,v}$? Are there many examples where the syntax is incorrect, but the keywords are? If these different subscores are highly correlated, one could imagine that the specific weighting of the different subscores matters very little.

### Description of VQA component.
- In section 2.1, there is a paragraph describing QA and fairness of the VQA pairs with human experts. What model is used to generate/conver the structured output for the "correct" subset of VQA pairs? My understanding is the the VLM judge (GPT-4.1-mini) is responsible for answering questions based on the rendered image. Is it also responsible for generating the image in this case?
- Questions which are judged unfair should plausibly be removed from the benchmark, but it is not clear that this is the case. Please make this clear.
- Please include a specific discussion of the benefits and drawbacks of VQA as an evaluation framework vs. evaluation of renderable programming artifacts.

### Accessibility.
- Please include a clear discussion of how practitioners can evaluate new LLMs on StructEval. I imagine that the paper will eventually link to a code repository, but special care should be taken to ensure that the need to work with different rendering environments (as discussed in the introduction) is not a barrier to use.

The following points would improve my evaluation of the paper, but are not critical to ensuring my recommendation.
### Error analysis
- How correlated are errors on individual samples across different LLMs? Within a certain task (e.g. $\text{Text}\rightarrow \text{TOML}$) are there much harder and easier examples which drive performance, or is there a great deal of variance in errors across LLMs?
### Minor points
- Second paragraph:"in these tasks" x2
- Figure 3 formatting: "Feature Requirements" should have a newline
- Please be careful of the use of \citet vs. \citep throughout the paper.
- In Table 6, it appears that delta is the difference between o1-min and Qwen3-4b. The paragraph **Open-Source vs. Closed-Source Models** describes the quantity $\Delta(\text{close}_{avg}-\text{open}_{avg})$. Is this meant  to be described in the table as well?
- Appendix A is titled "Example Appedix".

---

> ### Author Response · Authors · 2025-10-28
> **Rebuttal by Authors (Part 1)**
>
> **Q1:**
> > One concern I have is that the number of examples within the benchmark is quite small, ... It would be useful to know how performance of (even a subset) of LLMs on larger text based benchmarks relates to performance on StructEval.
>
> > SoEval is quite similar to the StructEval-T generation subdataset (without the requirement of flat schemas). ... I believe such an analysis would be quite useful to strengthen the case for StructEval as a general-purpose benchmark for structured data.
>
> Thank you for the comment. StructEval differs from previous benchmarks such as StrucText-Eval (Gu et al., 2024) and SoEval (Liu et al., 2024) in both **scope** and **coverage**.
>
> StrucText-Eval focuses on reasoning and understanding within structure-rich text (e.g., JSON, XML, Markdown, and LaTeX) and contains approximately 5,800 examples. SoEval, which includes about 3,700 examples, evaluates model ability to generate structured textual outputs such as JSON or XML, but all tasks are constrained to flat schemas without nested or visual structures.
>
> In contrast, StructEval covers 18 structured formats and 44 distinct task types, spanning both text-based (StructEval-T) and renderable (StructEval-V) outputs, including formats such as HTML, React, LaTeX, and SVG. While each subtask in StructEval contains around 50 examples, the benchmark has 2,035 examples in total, which is similar to other multi-format benchmarks. **Because StructEval includes generation tasks and cross-format conversion tasks, direct comparison with in dataset size is not equivalent. **
>
> StructEval evaluates LLMs through a unified framework that measures **syntax validity**, **structural fidelity**, and **visual-semantic alignment**, whereas StrucText-Eval and SoEval primarily assess correctness of **textual structures**. StructEval introduces new evaluation dimensions, such as layout and visual correctness, which is not included in previous benchmarks. Therefore, StructEval complements rather than replaces prior large-scale text-only datasets.
>
> We have computed correlation between StructEval-T generation results and StrucText-Eval Table 2 (RougeL “Base”) scores, which both measure structured text-generation performance. The two overlapping models are Llama-3.1-8B-Instruct and Qwen2-7B. Their results are shown below:
>
> | Model               | StructEval-T Generation | StrucText-Eval (RougeL Base) |
> |----------------------|------------------------:|------------------------------:|
> | Llama-3.1-8B-Instruct | 60.22 | 43.9 |
> | Qwen2-7B              | 59.21 | 70.4 |
>
> The computed Pearson correlation is **r = −1.00** and Spearman rank correlation is **ρ ≈ −1.00**, but this value is not meaningful due to the very small overlap (n = 2). Both datasets evaluate structured-generation ability, but StructEval-T spans a broader range of formats including JSON, YAML, XML, , while StrucText-Eval is restricted to text-only structured outputs evaluated by RougeL similarity. The inverse correlation reflects small-sample noise, not an actual trend.
>
> We have also computed correlation between StructEval-T generation results and StrucText-Eval Table 3 (Base) scores, which correspond to performance on the StrucText-Eval-Hard subset. This subset measures structured-text understanding and reasoning, rather than generation. The results are shown below:
>
> | Model               | StructEval-T Generation | StrucText-Eval (Hard Base) |
> |----------------------|------------------------:|----------------------------:|
> | Llama-3.1-8B-Instruct | 60.22 | 22.3 |
> | Gemini-1.5-Pro        | 88.07 | 11.2 |
> | GPT-4o-mini           | 79.86 | 39.3 |
> | Qwen2.5-7B-Instruct   | 59.21 | 29.6 |
>
> The computed Pearson correlation is **r = −0.27** and Spearman rank correlation is **ρ = −0.40**. The weak and negative correlation indicates that model performance on text-understanding tasks (StrucText-Eval-Hard) does not align directly with performance on structured text-generation tasks (StructEval-T). StructEval evaluates syntactic correctness and schema compliance in generated outputs, while StrucText-Eval-Hard measures reasoning over existing structured text, which explains the difference in trends.

---

> > ### Author Response · Authors · 2025-10-28
> > **Rebuttal by Authors (Part 2)**
> >
> > **Q2:**
> > > Equations describe one possible weighting of to generate a single score. This choice of weighting appears arbitrary, and downweights syntactic errors relative to others. One could imagine cases where syntactic errors are much more important than keyword errors, and it is not clear if the conclusions of the experiment would hold if a different scoring were to be used.
> >
> > Thanks for the suggestion, we have provided a few more differnt weight combinations of and present the results.
> >
> > ---
> >
> > ### Reweighting Configuration 1
> >
> > | Subset | Render Weight | Keyword Matching Weight | Syntax Weight | VQA Weight |
> > |:--|:--:|:--:|:--:|:--:|
> > | **StructEval-V** | 0.0 | 0.5 | — | 0.5 |
> > | **StructEval-T** | 0.5 | — | 0.5 | — |
> >
> > The results are:
> >
> > | Model                    | StructEval-V Avg | StructEval-T Avg | Overall Avg |
> > |----------------------------|:----------------:|:----------------:|:------------:|
> > | gpt-4o                   | 0.74 | 0.86 | **0.80** |
> > | o1-mini                  | 0.71 | 0.89 | **0.80** |
> > | gpt-4.1-mini             | 0.72 | 0.87 | 0.79 |
> > | gpt-4o-mini              | 0.72 | 0.84 | 0.78 |
> > | gemini-1.5-pro           | 0.68 | 0.85 | 0.76 |
> > | Qwen3-4B                 | 0.66 | 0.83 | 0.74 |
> > | gemini-2.0-flash         | 0.63 | 0.79 | 0.71 |
> > | Llama-3.1-8B-Instruct    | 0.63 | 0.75 | 0.69 |
> > | Phi-4-mini-instruct      | 0.59 | 0.75 | 0.67 |
> > | Qwen2.5-7B-Instruct      | 0.64 | 0.67 | 0.65 |
> > | Meta-Llama-3-8B-Instruct | 0.59 | 0.62 | 0.60 |
> > | Phi-3-mini-128k-instruct | 0.45 | 0.43 | 0.44 |
> >
> > ---
> >
> > ### Reweighting Configuration 2
> >
> > | Subset           | Render Weight | Keyword Matching Weight | Syntax Weight | VQA Weight |
> > |:-----------------|:-------------:|:------------------------:|:-------------:|:----------:|
> > | **Renderable**   |     0.5       |          0.5            |       —       |    0.0     |
> > | **Non-Renderable** |     0.5     |           —             |      0.5      |     —      |
> >
> > The results are:
> >
> > Model                    | Renderable Avg | Non-renderable Avg | Overall Avg
> > -|-|-|-
> > gpt-4o                   |           0.89 |               0.86 |        0.88
> > o1-mini                  |           0.87 |               0.89 |        0.88
> > gpt-4.1-mini             |           0.88 |               0.87 |        0.87
> > gemini-1.5-pro           |           0.88 |               0.85 |        0.87
> > gpt-4o-mini              |           0.88 |               0.84 |        0.86
> > Qwen3-4B                 |           0.86 |               0.83 |        0.84
> > gemini-2.0-flash         |           0.86 |               0.79 |        0.83
> > Llama-3.1-8B-Instruct    |           0.83 |               0.75 |        0.79
> > Phi-4-mini-instruct      |           0.82 |               0.75 |        0.79
> > Qwen2.5-7B-Instruct      |           0.84 |               0.67 |        0.76
> > Meta-Llama-3-8B-Instruct |           0.81 |               0.62 |        0.72
> > Phi-3-mini-128k-instruct |           0.80 |               0.43 |        0.63
> >
> > ---
> >
> > ### Reweighting Configuration 3
> >
> > | Subset           | Render Weight | Keyword Matching Weight | Syntax Weight | VQA Weight |
> > |:-----------------|:-------------:|:------------------------:|:-------------:|:----------:|
> > | **Renderable**   |     0.5       |          0.0            |       —       |    0.5     |
> > | **Non-Renderable** |     0.5     |           —             |      0.5      |     —      |
> >
> > The results are:
> >
> > Model                    | Renderable Avg | Non-renderable Avg | Overall Avg
> > -|-|-|-
> > gpt-4o                   |           0.74 |               0.86 |        0.80
> > o1-mini                  |           0.71 |               0.89 |        0.80
> > gpt-4.1-mini             |           0.72 |               0.87 |        0.79
> > gpt-4o-mini              |           0.73 |               0.84 |        0.78
> > gemini-1.5-pro           |           0.69 |               0.85 |        0.76
> > Qwen3-4B                 |           0.67 |               0.83 |        0.74
> > gemini-2.0-flash         |           0.62 |               0.79 |        0.70
> > Llama-3.1-8B-Instruct    |           0.64 |               0.75 |        0.69
> > Phi-4-mini-instruct      |           0.61 |               0.75 |        0.67
> > Qwen2.5-7B-Instruct      |           0.64 |               0.67 |        0.66
> > Meta-Llama-3-8B-Instruct |           0.58 |               0.62 |        0.60
> > Phi-3-mini-128k-instruct |           0.46 |               0.43 |        0.45
> >
> > ---
> >
> > We can see the ranking of these models generlly remain the same, except a few models has varying ranks under differnt weighting combinations.

---

> > > ### Author Response · Authors · 2025-10-28
> > > **Rebuttal by Authors (Part 3)**
> > >
> > > **Q3:**
> > >
> > > > As stated in the introduction, one of the main challenges of designing a benchmark like StructEval is building a framework which can support evaluation across rendering environments. ... it is especially important to explicitly address the question of how StructEval can be used in the main text of the paper.
> > >
> > > > Please include a clear discussion of how practitioners can evaluate new LLMs on StructEval. I imagine that the paper will eventually link to a code repository, but special care should be taken to ensure that the need to work with different rendering environments (as discussed in the introduction) is not a barrier to use.
> > >
> > > Thank you for your comment. StructEval follows a three-stage evaluation pipeline: inference, rendering, and evaluation.
> > >
> > > 1. In the inference stage, all prompts from the benchmark are executed using the LLM-Engine framework, which supports both open-source model backends (such as vLLM, SGLang, and Together) and commercial APIs (such as OpenAI, Claude, and Gemini). The generated outputs are enclosed between `<|BEGIN_CODE|>` and `<|END_CODE|>` tags to enable consistent parsing.
> > >
> > > 2. In the rendering stage, for StructEval-V tasks, the parsed structured outputs (such as HTML, LaTeX, SVG, React, Mermaid, or Matplotlib code) are processed by a unified rendering engine within a **pre-configured virtual environment containing all required packages**. The engine executes each format using deterministic back-ends, for instance, HTML codes are rendered through playwright.  For StructEval-T, the generated code is extracted into a disk file and is validated that it parses, later **the evaluator loads that file into a Python dictionary/list structure and checks required paths via dictionary traversal.** All rendering runs under the same environment to ensure consistent visual outputs and reproducible evaluation across different models.
> > >
> > > 3. In the evaluation stage, the benchmark automatically computes task-specific metrics: (1) Syntax Score, which checks structural validity using format-specific parsers or by verifying successful rendering; (2) Keyword Matching Score, which measures the proportion of expected structural elements using dot-path rules for hierarchical formats; and (3) VQA Score, used only for visual tasks, where GPT-4.1-mini evaluates whether visual question–answer pairs align with the rendered image. Final task scores are computed as weighted combinations of these metrics depending on whether the task is text-based or renderable.
> > >
> > > The full inference, rendering, and evaluation code will be released with the benchmark.
> > >
> > >
> > > **Q4:**
> > >
> > > > How correlated are subscores? Are there many examples where the syntax is incorrect, but the keywords are? If these different subscores are highly correlated, one could imagine that the specific weighting of the different subscores matters very little.
> > >
> > > The syntax score and keyword matching score measure different aspects of the output. The syntax score verifies whether the generated code is syntactically valid and can be successfully parsed or rendered. The keyword matching score checks whether all required fields specified in the task are present in the generated output. **It is possible for the syntax score to be 1 while the keyword matching score is 0 if the code is syntactically correct but missing required fields.** The keyword matching score is only computed when the syntax score equals 1.  if the syntax check fails, the keyword score is automatically set to 0.
> > >
> > > **Q5:**
> > > > In section 2.1, there is a paragraph describing QA and fairness of the VQA pairs with human experts. What model is used to generate/conver the structured output for the "correct" subset of VQA pairs? My understanding is the the VLM judge (GPT-4.1-mini) is responsible for answering questions based on the rendered image. Is it also responsible for generating the image in this case?
> > >
> > > Thank you for the questions. We confirm that the benchmark data was synthesized using o1-mini for StructEval-V (keywords and VQA pairs) and o3-mini for StructEval-T (dot-path verification), as described in our annotation pipeline. There seems to be a misunderstadning in how the image is rendered.  **Please note that the images are not generated using any models, but they are rendered deterministically.**
> > >
> > > For StructEval-V tasks, the parsed structured outputs (such as HTML, LaTeX, SVG, React, Mermaid, or Matplotlib code) are processed by a unified rendering engine within a pre-configured virtual environment containing all required packages. The engine executes each format using deterministic back-ends, for instance, HTML codes are rendered through playwright.

---

> ### Author Response · Authors · 2025-10-28
> **Rebuttal by Authors (Part 4.1)**
>
> **Q6:**
> > Questions which are judged unfair should plausibly be removed from the benchmark, but it is not clear that this is the case. Please make this clear.
>
> We have observed that there are two types of VQA questions, **fair VQA** that can be reasonably answered by the VLM judge when given the one rendered image, and **unfair VQA** that cannot be reasonably answered by the VLM judge.  For example, questions that require precise numerical values or interactive components that cannot be visually verified (e.g. asking the exact size of the image, or interactive components based on one rendered image, or the exact color number). The unfair questions were later refined by our annotators to ensure that they can be reasonably answered by our VLM judge.
>
> We will make this more clear in the final version of the paper.
>
>
> **Q7:**
> > Please include a specific discussion of the benefits and drawbacks of VQA as an evaluation framework vs. evaluation of renderable programming artifacts.
>
> For StructEval-V, **we do not have a single ground-truth rendered image because each task can produce multiple visually equivalent outputs that all satisfy the same structural constraints.** Therefore, direct pixel-level or layout-level comparison is not feasible. **Instead, we evaluate rendered outputs using VQA pairs that check whether key visual attributes (e.g., text content, element presence, or layout relationship) are satisfied.** This approach allows flexible yet objective evaluation of visual correctness. The drawback is that the evaluation depends on the accuracy of the vision-language model (GPT-4.1-mini) used as the judge; we assume it answers correctly given the rendered image and the question–answer pairs.
>
> *Q8:**
> > How correlated are errors on individual samples across different LLMs? Within a certain task (e.g. ) are there much harder and easier examples which drive performance, or is there a great deal of variance in errors across LLMs?
>
> Thank you for your valuable suggestion, we have calculated the correlations on the errors on individual samples across different LLMs, and here is a table:
>
> ### Selected pairwise error correlations
> | Pair | Correlation |
> |---|---|
> | Llama-3.1-8B-Instruct ↔ Phi-3-mini-128k-instruct | 0.67 |
> | Meta-Llama-3-8B-Instruct ↔ Phi-3-mini-128k-instruct | 0.66 |
> | Llama-3.1-8B-Instruct ↔ Meta-Llama-3-8B-Instruct | 0.63 |
> | Phi-4-mini-instruct ↔ gemini-2.0-flash | 0.60 |
> | Qwen2.5-7B-Instruct ↔ o1-mini | 0.54 |
> | gpt-4o ↔ o1-mini | 0.49 |
> | Llama-3.1-8B-Instruct ↔ gpt-4.1-mini | 0.27 |
> | Meta-Llama-3-8B-Instruct ↔ gpt-4.1-mini | 0.28 |
> | Phi-3-mini-128k-instruct ↔ gpt-4.1-mini | 0.24 |
> | gemini-1.5-pro ↔ gpt-4.1-mini | 0.14 |
>
>
> The first table reports how similarly pairs of models succeed or fail on the same items. For each pair, we take all items that both models attempted and convert each model’s score on an item into a binary “correct/incorrect” using a 0.5 threshold. We then compute the Pearson correlation between the two binary vectors. A correlation of 1.00 (the diagonal) means a model is perfectly correlated with itself.
>
> Off‑diagonal values quantify cross‑model alignment of errors: for example, 0.67 for Llama‑3.1‑8B‑Instruct versus Phi‑3‑mini‑128k‑instruct indicates these two models tend to get the same items right or wrong much of the time, whereas 0.14 for gemini‑1.5‑pro versus gpt‑4.1‑mini indicates little shared pattern in which items they miss. Values near 0.6 (e.g., Llama‑3.1‑8B‑Instruct with Meta‑Llama‑3‑8B‑Instruct at 0.63; Phi‑4‑mini‑instruct with gemini‑2.0‑flash at 0.60) show alignment, while values around 0.24–0.29 for several pairs with gpt‑4.1‑mini show weak alignment.
>
> (continue in next block)

---

> ### Author Response · Authors · 2025-10-28
> **Rebuttal by Authors (Part 4.2)**
>
> (continue from part 4.1)
>
> ### Exemplars of item difficulty and variance
>
> | TaskID | Mean | Std | Median | Acc=1 | >0.8 | 0.6-0.8 | 0.4-0.6 | 0.2-0.4 | 0-0.2 | =0 |
> |---|---:|---:|---:|---:|---:|---:|---:|---:|---:|---:|
> | 050229(JSON -> CSV) | 1.00 | 0.00 | 1.00 | 12 | 12 | 0 | 0 | 0 | 0 | 0 |
> | 021716(CSV -> XML) | 0.12 | 0.10 | 0.20 | 0 | 0 | 0 | 0 | 7 | 11 | 5 |
> | 021726(CSV -> XML) | 0.18 | 0.06 | 0.20 | 0 | 0 | 0 | 0 | 11 | 0 | 1 |
> | 021745(CSV -> XML) | 0.17 | 0.08 | 0.20 | 0 | 0 | 0 | 0 | 10 | 0 | 2 |
> | 001047(Text -> TOML) | 0.50 | 0.52 | 0.48 | 5 | 6 | 0 | 0 | 0 | 0 | 6 |
> | 171845(XML to YAML) | 0.57 | 0.46 | 0.60 | 6 | 6 | 0 | 0 | 4 | 0 | 2 |
> | 021847(CSV to YAML) | 0.72 | 0.45 | 1.00 | 8 | 8 | 1 | 0 | 0 | 0 | 3 |
>
> The second table summarizes, for each sampled task, how the 12 models distributed their scores. “Mean,” “Std,” and “Median” are computed across models for that task. The count columns partition model scores into ranges: “Acc=1” counts models that achieved exactly 1.0, “>0.8,” “0.6–0.8,” “0.4–0.6,” “0.2–0.4,” “0–0.2,” and “=0” count how many model scores fell in each bin.
>
> Reading the examples:
> - task 050229 has mean 1.00 and std 0.00 with all 12 models in “Acc=1,” so it is uniformly solved.
> - Tasks 021716 (mean 0.12, std 0.10) and 021726 (mean 0.18, std 0.06) have most model scores in the lowest bins, indicating they are difficult.
> - Task 001047 (mean 0.50, std 0.52) shows a bimodal pattern, five models scored exactly 1.0 and six scored exactly 0—so models disagree sharply on that item.
> - Tasks like 021847 (mean 0.72, std 0.45) combine many perfect scores (eight at 1.0) with several zeros (three at 0), that some tasks show both strong successes and failures across models.
> Together, the correlation table shows that some model pairs share error patterns much more than others, and the per‑task table shows that the dataset contains uniformly easy items, broadly hard items, and items where model performance varies widely.
>
> Across the 12 models in the analysis, binary-correctness correlations on overlapping items span 0.14 to 0.67. The strongest off‑diagonal alignment is between Llama-3.1-8B-Instruct and Phi-3-mini-128k-instruct at 0.67; other high alignments include Llama-3.1-8B-Instruct with Meta-Llama-3-8B-Instruct at 0.63 and Phi-4-mini-instruct with gemini-2.0-flash at 0.60. At the low end, gemini-1.5-pro with gpt-4.1-mini is 0.14, and several pairs involving gpt-4.1-mini are between 0.24 and 0.29. These values quantify that some model pairs tend to succeed and fail on the same items considerably more than others, while certain pairs show much weaker alignment of errors.
>
> Within the 50 sampled items, **there are clear examples of both easy and hard items, as well as items with substantial cross‑model variance.**
> - Task 050229 is unanimously correct across all 12 models (mean 1.00, standard deviation 0.00, Acc=1 count 12). In contrast
> - Tasks 021716 (mean 0.12, std 0.10, with 11 model scores in the 0–0.2 range and 5 exact zeros), 021726 (mean 0.18, std 0.06, with 11 scores in 0.2–0.4 and 1 exact zero), and 021745 (mean 0.17, std 0.08, with 10 scores in 0.2–0.4 and 2 exact zeros) are predominantly hard.
>
> There are also high‑variance items where models disagree markedly:
> - task 001047 has mean 0.50 and std 0.52 with five exact 1.0s and six exact 0s across models; task 171845 has mean 0.57 and std 0.46 with six exact 1.0s and two exact 0s;
> - task 021847 has mean 0.72 and std 0.45 with eight exact 1.0s and three exact 0s.
>
> These figures show that, in the evaluated sample, both distinctly easy and distinctly hard examples exist and, for a subset of items, error patterns vary substantially across LLMs.
>
>
> **Q9:**
> > Second paragraph:"in these tasks" x2
>
> > Figure 3 formatting: "Feature Requirements" should have a newline
>
> > Please be careful of the use of \citet vs. \citep throughout the paper.
>
> > In Table 6, it appears that delta is the difference between o1-min and Qwen3-4b. The paragraph Open-Source vs. Closed-Source Models describes the quantity $\Delta(\text{close}{avg}-\text{open}{avg})$. Is this meant to be described in the table as well?
>
> > Appendix A is titled "Example Appedix".
>
> Thank you for your valuable suggestions, we will edit and update our manuscript.
>
> **Q10.**
>
> > Given that evaluation of large language models on structured data is immediately relevant to many large scale industrial applications, I believe that a broader impact statement around the implications of this work for (for example) data privacy, climate, and fairness would be useful.
>
> Thank you for your suggestion, we will add a section explaining the implication of this work for data privacy, climate and fairness, etc, later.

---

### Review · Reviewer_hMWu · 2025-10-09

**Summary Of Contributions:**

The paper proposes StructEval, a benchmark for LLM structured data generation (e.g., JSON, HTML). Previous benchmarks focused more on LLM reasoning capabilities and the semantic quality of the generated outputs without much emphasis on whether the generated output conforms to the required format. The authors argue the importance of this benchmark as LLM is being increasingly used in software development. State-of-the-art models like o1-mini still produce subpar performance on this benchmark, and open source counterparts lagging 10 points behind.

**Additional Comments:**

N/A

**Audience:**

Yes

**Audience Explanation:**

I am convinced by the importance of the benchmark: LLM must be able to accurately generate structured outputs in order to assist in software engineering.

**Claims And Evidence:**

Yes

**Claims Explanation:**

The paper provides appropriate and convincing evidence regarding the following aspects of the benchmark:
1. Data curation
2. Evaluation criteria and metric design
3. Annotation pipeline
4. Model evaluation on the benchmark

**Requested Changes:**

minor in writing:
- intro paragraph 2 line 4 “in these tasks” repeated twice

requested additional details and experiments:
- how was the human experts for VQA questions evaluation gathered? can we have more details about the evaluators?
- If budget permits, would it be possible to add currect SOTA models from GPT series and Claude series?

---

> ### Author Response · Authors · 2025-10-28
> **Rebuttal by Authors**
>
> **Q1:**
> > intro paragraph 2 line 4 “in these tasks” repeated twice
>
> Thank you for your suggestion, we will edit the typo and update the manuscript.
>
>
> **Q2:**
> > how was the human experts for VQA questions evaluation gathered? can we have more details about the evaluators?
>
> They are mostly graduate level computer science researchers.
>
>
> **Q3:**
> > If budget permits, would it be possible to add currect SOTA models from GPT series and Claude series?
>
> Thank you for your valuable suggestion. However, due to high cost and budget limit, we are not able to add these models at the moment, but we will provide reproducible code that includes a LLM harness that can run these models.
>
> We added a newer model from gemini, here is the result for gemini-2.5-flash
>
> |Models | StructEval-T (generation) | StructEval-T (conversion) | StructEval-V (generation) | StructEval-V (conversion) | Average |
> -|-|-|-|-|-
> gemini-2.5-flash | 91.06 | 78.84 | 67.96 | 65.33 | 76.55

---

### Decision · Action_Editor_daiY · 2025-12-07

**Recommendation:** Accept as is

**Additional Comments:**

This paper proposes a benchmark to evaluate LLM capabilities in producing various forms of structured data. Achieving this capability can be challenging, because models must be able to handle the constraints of structured generation in addition to generating reasonable content. The authors propose a diverse benchmark (with a number of formats, tasks, and metrics(). The benchmark is not yet saturated and is likely to be very useful to model builders and practitioners.

The reviewers have asked a bunch of clarifying questions. The authors have produced convincing responses and made suitable updates. Overall I think this is a valuable work for the community and should be accepted.

**Audience:**

Yes

**Audience Explanation:**

Yes, this work provides new insights into the capabilities of LLMs on a very common form of data.

**Claims And Evidence:**

Yes

**Claims Explanation:**

Yes, the authors have a strong methodology and provide solid evidence.